# Evidence for Naturally Produced Beauvericins Containing *N*-Methyl-Tyrosine in *Hypocreales* Fungi

**DOI:** 10.3390/toxins11030182

**Published:** 2019-03-26

**Authors:** Monika Urbaniak, Łukasz Stępień, Silvio Uhlig

**Affiliations:** 1Plant-Pathogen Interaction Team, Department of Pathogen Genetics and Plant Resistance, Institute of Plant Genetics of the Polish Academy of Sciences, Strzeszyńska 34, 60-479 Poznań, Poland; lste@igr.poznan.pl; 2Norwegian Veterinary Institute, P.O. Box 750 Sentrum, 0106 Oslo, Norway; silvio.uhlig@vetinst.no

**Keywords:** beauvericin, beauvenniatin, depsipeptide, *Fusarium*, *Isaria*, *Paecilomyces*, mycotoxin

## Abstract

Beauvericin is a depsipeptide mycotoxin. The production of several beauvericin analogues has previously been shown among various genera among *Hypocreales* fungi. This includes so-called beauvenniatins, in which one or more *N*-methyl-phenylalanine residues is exchanged with other amino acids. In addition, a range of “unnatural” beauvericins has been prepared by a precursor addition to growth medium. Our aim was to get insight into the natural production of beauvericin analogues among different *Hypocreales* fungi, such as *Fusarium* and *Isaria* spp. In addition to beauvericin, we tentatively identified six earlier described analogues in the extracts; these were beauvericin A and/or its structural isomer beauvericin F, beauvericin C, beauvericin J, beauvericin D, and beauvenniatin A. Other analogues contained at least one additional oxygen atom. We show that the additional oxygen atom(s) were due to the presence of one to three *N*-methyl-tyrosine moieties in the depsipeptide molecules by using different liquid chromatography–mass spectrometry-based approaches. In addition, we also tentatively identified a beauvenniatin that contained *N*-methyl-leucine, which we named beauvenniatin L. This compound has not been reported before. Our data show that *N*-methyl-tyrosine containing beauvericins may be among the major naturally produced analogues in certain fungal strains.

## 1. Introduction

Hexadepsipeptides are non-ribosomal cyclic metabolites from fungi. The beauvericins and beauvenniatins are structurally and genetically related compounds, and they belong in this group. Beauvericins consist of three alternating aromatic *N*-methyl-L-amino acid moieties and up to three D-2-hydroxyisovaleric (D-Hiv) acid moieties [1,2]. The thus far identified four analogues of beauvericin (A, B, C, and F) contain one, two, or three groups of 2-hydroxyisocaproic acid (D-Hmp) instead of D-Hiv, while the amino acid is *N*-methyl-phenylalanine. To date, five naturally occurring beauvericin analogues have been identified, as well as 11 analogues from precursor-directed synthesis [3,4,5,6]. Beauvericins are produced by a wide range of *Hypocreales* fungi, such as phytopathogenic *Fusarium* species or entomopathogenic *Beauveria* and *Isaria* (syn. *Paecilomyces*) species [7,8,9]. Beauvericins have inhibitory activity towards *Mycobacterium tuberculosis* or *Plasmodium falciparum*, and are cholesterol acyltransferase inhibitors [3,10]. Furthermore, they have shown antifungal activities, suggesting their potential use as co-drugs for antifungal infections in humans [11]. Beauvericins displayed relatively strong cytotoxicity in different human cancer cell lines, likely in part because of their ionophoric properties causing non-specific membrane disruption in vitro [3,12,13].

Beauvenniatins contain one to two amino acids that are commonly found in enniatins, e.g., *N*-methyl-valine, *N*-methyl-leucine, or *N*-methyl-isoleucine, in addition to *N*-methyl-phenylalanine [14,15]. Beauvenniatins A-F, G_1_, G_2_, G_3_, H_1_, H_2_, and H_3_ were detected in an *Acremonium* sp. mycelial extract. Beauvenniatins F, G_1_, G_2_, G_3_, H_1_, H_2_, and H_3_ contain one *N*-methyl-L-phenylalanine and two *N*-methyl-L-valine residues, the difference between the analogues being variations in their hydroxy acid composition [14,15]. Several beauvenniatins have only been shown as products from precursor-directed biosynthesis, i.e., addition of the corresponding amino acid precursor to the growing medium was necessary for their production. Beauvenniatins C, D, and E are *N*-methyl-tyrosine containing analogues that had been identified in a culture of an *Acremonium* sp. after the addition of L-tyrosine to the growth medium. Beauvenniatins A and B exhibit similar bioactivities as beauvericin against *Mycobacterium tuberculosis*, *Plasmodium falciparum*, cancer cell lines (KB, MCF-7, and NCl-H187) and nonmalignant Vero cells. Beauvenniatins containing *N*-methyl-tyrosine residues were relatively less bioactive or inactive [14,15].

The number of reported analogues that belong into the beauvericin and beauvenniatin group has increased substantially in the last decade. This indicates that there may exist a high number of beauvericins and beauvenniatins that are naturally produced by fungi. Depsipeptides such as beauvericin, beauvenniatins and enniatins possess remarkable biological activities; they may be toxic and might co-occur with other mycotoxins. They occur in crops, vegetables, grains and grain-based foodstuffs, and may be concentrated during food production and processing [8,16,17,18,19].

In this study, we demonstrate naturally produced *N*-methyl-tyrosine containing beauvericin analogues produced by *Fusarium* and *Isaria* species. Analogues of corresponding composition have previously been shown from precursor-directed synthesis. Moreover, we show the high variability of beauvericin-type cyclohexadepsipeptides in different *Hypocreales* fungi.

## 2. Results and Discussion

Fungal strains were identified using sequencing of the PCR-amplified specific genomic region. The DNA fragments were amplified with ITS4/ITS5 and Ef728M/TefR1 primers, sequenced, and compared with reference sequences deposited in the GenBank Database. The complete sequences of the amplified internal transcribed spacers (ITS) regions from the fungal strains indicated over 99% identity to individual sequences. Results from molecular identification are described in Table 1.

During our LC–MS based metabolite screening of extracts from rice cultures of *Fusarium* spp., the strains RT 6.7, RT 5.4, MU12, P35, and PIN 5.5, as well as *Isaria farinosa* 4447 contained a large number of different beauvericin and beauvenniatin analogues (Table 1, Figure 1 and Figure 2). In reversed-phase liquid chromatography, the elution order of different beauvericins and beauvenniatins was strongly correlated to their molecular weight. The putative beauvericins and beauvenniatins afforded abundant [M + H]^+^, [M + NH_4_]^+^, and [M + Na]^+^ ions during electrospray ionization. In total, eight beauvericins (**1**–**8**) and two beauvenniatins (**9**, **13**) were detected in the different fungal extracts (Table 1). All 10 depsipeptides, except **13**, had been reported earlier (Figure 1) [3,5,6,14,20]. However, several of the analogues were only reported from precursor-directed biosynthesis [3,5,6,14,20].

### 2.1. Mass Spectrometry and Tentative Structure Determination of Beauvericins

Beauvericin (**1**) was the major analogue in all of the fungal extracts that contained beauvericin analogues. A common feature of all beauvericins is that they contain three *N*-methyl-phenylalanine moieties, which in some cases are partly exchanged by phenylalanine. Thus, the beauvericins we tentatively identified in the seven fungal strains containing three *N*-methylated amino acids, while in **4**, one of the phenylalanine moieties is not *N*-methylated. We used liquid chromatography interfaced to high-resolution mass spectrometry (LC–HRMS) to establish the elemental compositions and peptide sequences of beauvericins in the *F. concentricum* and *I. farinosa* strains, since these two strains together contained all of the detected beauvericin analogues (Table 2, Figure 2). Beauvericins comprising three *N*-methyl-phenylalanine or phenylalanine moieties contain nine oxygen atoms (Figure 1, Table 2). We detected three beauvericins with a larger number of oxygen atoms, i.e., **6**, **7**, and **8** contained 10, 11, or 12 oxygen atoms, respectively (Table 2). The elemental composition of three other beauvericins was consistent with **2/5**, **3**, and **4** (Table 2).

The [M + Na]^+^ ions of beauvericins and related depsipeptides have previously been shown to yield product ions of high diagnostic value for structure determination [3,14]. Thus, fragmentation of the sodiated molecular ions allows sequencing of the depsipeptides, since the molecules primarily fragment at the amide and ester bonds [5,15]. We used **1** as a model molecule for comparing the fragmentation patterns with the putative beauvericin analogues. Since **1** consists exclusively of *N*-methyl-phenylalanine and D-2-hydroxyisovaleric (D-Hiv) moieties, the HRMS/MS spectrum from higher-energy collision dissociation (HCD) was a superimposition of a series of subsequent cleavages attributed to loss of −161 Da and −100 Da due to *N*-methyl-phenylalanine and D-Hiv, respectively (Figure 3). The product ion spectra of **6** and **7** traced the oxygenation to the amino acid moiety, i.e., the −161 Da product ions from loss of *N*-methyl-phenylalanine were accompanied by −177 Da product ions (Figure 3). The amino acid tyrosine is equivalent to 4-hydroxy-phenylalanine; thus, we hypothesized that the additional oxygen in **6** and **7** was due to the partial exchange of *N*-methyl-phenylalanine by *N*-methyl-tyrosine. Furthermore, the relative ratio between the [M − 161 + Na]^+^ and [M − 177 + Na]^+^ ions was approximately 2:1 in 6 and 1:2 in 7, indicating that the former contained one *N*-methyl-tyrosine residue, and the latter contained two. Compound 8 was the analogue that was produced in the smallest relative amounts, and the HRMS/MS spectra of its sodiated ions were of poor signal/noise (Appendix A).

In order to verify the identity of the amino acid *N*-methyl-tyrosine in **5**, **6**, and **7**, we aimed to acid hydrolyze the molecules and compare the amino acid composition to authentic standards. We successfully pre-separated the *F. concentricum* and *I. farinosa* extracts using solid-phase extraction to yield a fraction that contained **6**, **7**, and **8** as major constituents (Table 3). Hydrophilic interaction chromatography (HILIC) coupled to ion trap mass spectrometry of the hydrolysates, in comparison with reference standards, showed the presence of both *N*-methyl-phenylalanine and *N*-methyl-tyrosine in the enriched fraction (Figure 4, Appendix A). Since **7** and **8** have not yet been named, we suggest they should be named beauvericin K and beauvericin L, respectively, since the mono-*N*-methyl-tyrosyl analogue has been named beauvericin J (Figure 1) [14].

The presence of *N*-methyl-tyrosine in beauvericins and beauvenniatins has previously been shown in an *Acremonium* sp. [14] In that study, beauvericin J (**6**) and beauvenniatins C (**10**), D (**11**), and E (**12**) were obtained from precursor-directed biosynthesis, and their structures were elucidated using NMR spectroscopy [14].

While most beauvericins contain exclusively D-Hiv, in compounds **2/5** and **3**, one or three D-Hiv moieties are exchanged with D-hydroxy-isocaproic acid (2-hydroxy-4-methylpentanoic acid, or D-Hmp) (Figure 1). This exchange is reflected in the HRMS/MS spectra of **2/5** and **3** (Appendix A). While the presence of D-Hiv gives fragments of −100 Da, the presence of D-Hmp yields fragments of −114 Da. For example, HRMS/MS of beauvericin A/F (**2/5**) gave *m/z* 559.2770 ions, which corresponds to the loss of one *N*-methyl-phenylalanyl moiety and one D-Hiv unit (Appendix A). In addition, the HRMS/MS spectrum showed the presence of *m/z* 545.2610 ions, which are due to the loss of one *N*-methyl-phenylalanyl moiety and one D-Hmp unit (Appendix A). In contrast, beauvericin C (**3**) showed a product ion sequence comprising of amino acid (*N*-methyl-phenylalanine, −161 Da) and hydroxy acid (D-Hmp, −114 Da) losses (Appendix A). We aimed also to determine the hydroxy acids in the hydrolysates, and D-Hiv was available as a reference standard. The compound afforded [M−H]^−^ ions at *m*/*z* 117 (Appendix A), while it did not ionize in positive ion mode (Appendix A). A reference standard for D-Hmp was not available, and we did not observe any ion of reasonable signal/noise for a possible D-Hmp in the hydrolysate.

### 2.2. Mass Spectrometry and Tentative Structure Determination of Beauvenniatins

Beauvenniatins can be distinguished from beauvericins by the presence of amino acid losses in the MS/MS spectra of the [M + Na]^+^ ions that are different from the characteristic −161 Da losses due to *N*-methyl-phenylalanine. Another characteristic is the lower number of carbon atoms and higher number of hydrogen atoms (and thus the lower number of ring double bond equivalents) compared to beauvericins due to the lack of the phenyl moiety (Table 2). Thus, beauvenniatin A (**9**) contains one group of *N*-methyl-valine in addition to two *N*-methyl-phenylalanine moieties, which is reflected in the presence of [M − 113 + Na]^+^ ions in addition to [M − 161 + Na]^+^ ions in the product ion spectra (Figure 5).

We detected another major depsipeptide that afforded [M + NH_4_]^+^ and [M + Na]^+^ ions with *m/z* 767.4583 and *m/z* 772.4104, respectively (Table 2). Fragmentation of the [M + Na]^+^ ions gave [M − 127 + Na]^+^ ions in addition to [M − 161 + Na]^+^ ions in the HRMS/MS product ion spectra (Figure 5). The loss of −127 Da in fungal depsipeptides is often due to the presence of *N*-methyl-leucine or *N*-methyl-isoleucine. However, the amino acid isomers cannot be distinguished by mass spectrometry alone. Therefore, we hydrolyzed the 2% methanol fraction from silica gel fractionation, and subsequently compared the amino acid mixture with reference standards (Table 2, Figure 4). While both *N*-methyl-leucine and *N*-methyl-isoleucine afforded [M + H]^+^ ions with *m/z* 146, the two isomers were chromatographically separated using HILIC (Figure 4). Using this approach, we were able to show that the amino acid in the beauvenniatin was *N*-methyl-leucine (Figure 4). Thus, the depsipeptide was equivalent to beauvenniatin L (**13**).

## 3. Conclusions

This study demonstrates the high variability of beauvericin-type cyclohexadepsipeptides in *Fusarium* and *Isaria* strains. It also emphasizes that these fungi may naturally produce new types of beauvericins, such as *N*-methyl-tyrosine containing analogues, as well as new types of beauvenniatins.

## 4. Materials and Methods 

### 4.1. Fungal Strains

*Fusarium* and *Isaria* species were isolated from natural environments and deposited in the KF pathogenic fungi collection of the Institute of Plant Genetics, Polish Academy of Sciences, Poznań, Poland (Table 1).

### 4.2. Media and Growth Conditions

Growing mycelia of individual *Fusarium* and *Isaria* strains were purified and maintained in cultures for seven days on potato dextrose agar medium (PDA, Oxoid) for genomic DNA extraction. Furthermore, individual *Fusarium* and *Isaria* strains were cultivated for 14 days using sterile rice grain cultures for qualitative and quantitative analyses of cyclohexadepsipeptides.

### 4.3. DNA Extraction and Molecular Identification of Fungal Strains

A modified method with the use of the CTAB (hexadecyltrimethylammonium bromide) was used for genomic DNA extraction, which was described earlier by Kozłowska et al. [21]. *Isaria* species identification was performed on the basis of the sequence analysis of the internal transcribed spacers of the ribosomal DNA region (ITS1–ITS2), while *Fusarium* species were identified on the basis of the sequence analysis of a variable fragment of the translation elongation factor 1α gene (tef-1α).

Polymerase chain reactions (PCRs) were performed as described earlier by Gálvez et al. [19] using DreamTaq Green DNA polymerase (Thermo Scientific, Espoo, Finland). For the PCR amplification of a partial region of the tef-1α gene primers Ef728M—forward primer (5′-CATCGAGAAGTTCGAGAAGG-3′) and TefR1—reverse primer (5′-GCCATCCTTGGAGATACCAGC-3′) [22,23,24] were used, whereas for the PCR amplification of the internal transcribed spacers of the ribosomal DNA region, the following primers were used: ITS4—forward primer (5′-TCCTCCGCTTATTGATATGC-3′) and ITS5—reverse primer (5′-GGAAGTAAAAGTCGTAACAAGG-3′) [21,25]. Amplicons were separated in 1.5% agarose gel (Invitrogen) with GelGreen Nucleic Acid Stain (Biotium, Inc.).

For sequence analysis, PCR-amplified DNA fragments were purified as described earlier by Kozłowska et al. [26], DNA fragments were labeled using forward primer and the BigDyeTerminator 3.1 kit (Applied Biosystems, Foster City, CA, USA), according to the producer’s recommendation, and precipitated with 96% ethanol. Sequence reading was performed using Applied Biosystems equipment. Sequences were analyzed using the BLASTn algorithm against the GenBank database-deposited reference sequences. 

### 4.4. Chemicals

The following reagents were from Sigma-Aldrich (St. Louis, MO, USA): ammonium formate (>99.99%), *N*-methyl-l-tyrosine (>98%), *N*-methyl-l-phenylalanine (>98%), *N*-methyl-l-isoleucine (>98%), *N*-methyl-l-leucine (>98%), *N*-methyl-l-valine (>98%), beauvericin (>99%), hydrochloric acid solution (1.0 N, BioReagent, suitable for cell culture), and sodium hydroxide solution (10 M in H_2_O, BioUltra, for molecular biology). The following reagents were from Fluka (Buchs, Switzerland): d-α-hydroxyisovaleric acid (>98%), L-α-hydroxyisovaleric acid (>98%), and ammonium carbonate (HPLC grade). Methanol and acetonitrile (both gradient quality) were from Romil (Cambridge, England), while chloroform (HPLC grade) was from Rathburn (Walkerburn, Scotland). Acetonitrile and water for LC–MS were from Fisher Scientific (LC–MS grade, Thermo Fisher Scientific, Waltham, MA, USA). 

Standards of amino acids and beauvericin were dissolved with 25 ml of MeOH/water (1:1, *v*/*v*). The working stock concentration of beauvericin was 1 mg/mL for *N*-methyl-l-phenylalanine, and 2 µg/mL for *N*-methyl-l-isoleucine, *N*-methyl-l-leucine, and *N*-methyl-l-valine, while it was 0.1 µg/mL for *N*-methyl-l-tyrosine. 

### 4.5. Extraction

The rice cultures were lyophilized and ground into powder. For LC–MS screening of fungal cultures, aliquots of each culture were weighed (0.15 g) and extracted using 1.5 mL of methanol/water (9:1, *v*/*v*) by shaking on an orbital shaker (225 min^−1^, 90 min) followed by sonication for 20 min. After centrifugation on a Beckman J2-MC centrifuge (Beckman Coulter Inc., Fullerton, CA) for 10 min and 15,000× *g*, all of the extracts were filtered through a 0.22-µm nylon membrane (Costar, Corning Inc., Corning, NY, USA) and transferred to HPLC vials for LC–MS analyzes. For chromatographic fractionation on silica gel, culture aliquots (0.1 g) were weighed into 1.5-mL Eppendorf tubes and extracted with 1 mL of chloroform by sonication for 15 min. Samples were centrifuged at 15,000× *g* for 10 min, and then applied to silica gel columns (see below).

### 4.6. Liquid Chromatography–Ion-Trap Mass Spectrometry (ITMS)

Analytical liquid chromatography of extracts was carried out on an LCQ Fleet ion trap mass spectrometer (Thermo Fisher Scientific) coupled to a Waters Acquity UPLC (Milford, MA, USA). Separation was achieved on a SunFire C_18_ column (Waters; 50 × 2.1 mm, 3.5 µm particles) with 0.5-µm pre-column filter (Supelco, Bellefonte, PA, USA). Elution proceeded by means of a linear gradient with a 0.25 mL/min flow rate utilizing mobile phases A (5 mM of ammonium formate) and B (solution 5 mM of ammonium formate in MeOH/water, 95:5, *v*/*v*) as follows: 0–0.5 min, 20% A, 80% B; 0.5–15 min, 20% A, 80% B; 15–18 min 100% B; 18–21 min, 20% A, 80% B. The instrument was run in the full-scan mode (*m/z* 200–1200), and the electrospray interface was operated in the positive ion mode. Automatic tuning of the tube lens offset and skimmer voltage was carried out using a solution of 10 µg/mL of beauvericin (solution MeOH/water, 9:1, *v*/*v*).

### 4.7. Liquid Chromatography High-Resolution Mass Spectrometry (HRMS)

In order to determine the elemental composition of individual compounds and acquire high-resolution product ion spectra, extracts were analyzed using a Q-Exactive Fourier-transform high-resolution mass spectrometer (Thermo Fisher Scientific), which was interfaced to a Dionex UltiMate 3000 UHPLC (Thermo Fisher Scientific). The chromatographic method was identical to the one used in connection with ITMS. The mass resolution of the Q-Exactive was set to 70,000 at *m/z* 200, and the isolation width of the quadrupole set to 2 Da around the target ion for MS/MS. Product ion spectra were recorded for nine metabolites by fragmentation of the sodiated molecular ions using higher-energy collision dissociation (HCD) using a normalized collision energy of 35%. The instrument was run in the full-scan mode (*m/z* 150–1200). Elemental compositions were calculated using Xcalibur, version 2.3. (Thermo Fisher Scientific).

### 4.8. Chromatographic Fractionation, Acid Hydrolysis, and Amino and Hydroxy Acid Analysis

Chloroform extracts were applied to 100-mg silica gel columns (Phenomenex, Torrance, CA, USA) that had previously been conditioned with 2 mL of chloroform. The columns were eluted with 2 mL portions of 1–10% methanol in chloroform. All of the fractions were evaporated to dryness at 60 °C using a gentle stream of nitrogen. Residues were dissolved in 200 µL of methanol/water (9:1, *v*/*v*) and analyzed using the LC–ITMS method. For acid hydrolysis, the appropriate fractions were evaporated and dissolved in 200 µL of acetonitrile, followed by the addition of 200 µL of 6 M of hydrochloric acid. The vials were placed at 100 °C for 16 h. After cooling to room temperature, the hydrolysates were analyzed using hydrophilic interaction chromatography (HILIC) interfaced to the LCQ Fleet ion trap mass spectrometer. HILIC was performed using a SeQuant ZIC-pHILIC polymeric column (150 × 4.6 mm i.d.; 5 µm; Merck KGaA, Darmstadt, Germany). Elution proceeded by means of a linear gradient with a 0.25 mL/min flow rate utilizing solvents A (20 mM of ammonium carbonate) and B (acetonitrile) as follows: 0–1 min, 80% B; 1–29 min, 20% B; 29–35 min, 8% B; 35–36 min, 80% B. The mass spectrometer was operated in the full-scan mode in. the mass range *m/z* 90–250.

## Figures and Tables

**Figure 1 toxins-11-00182-f001:**
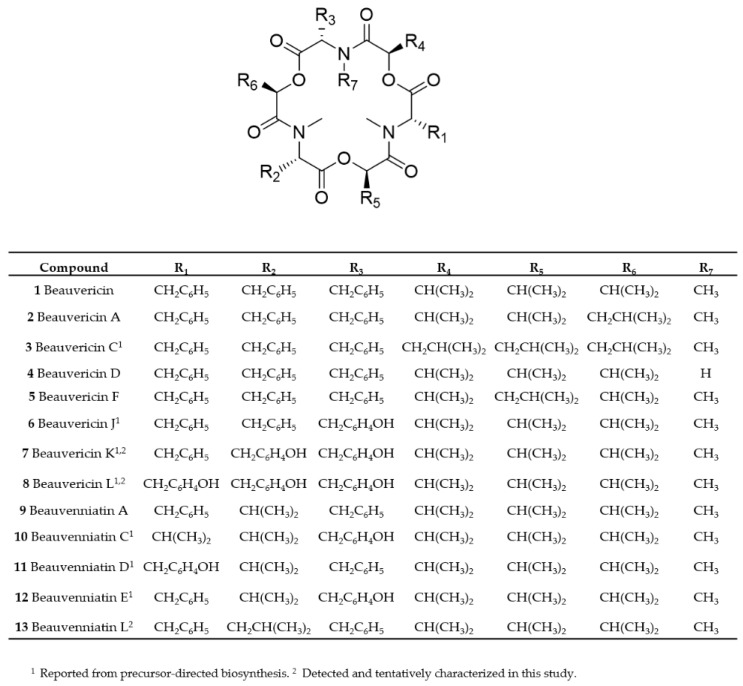
Overview of chemical structures of beauvericin and beauvenniatin analogues discussed in this study.

**Figure 2 toxins-11-00182-f002:**
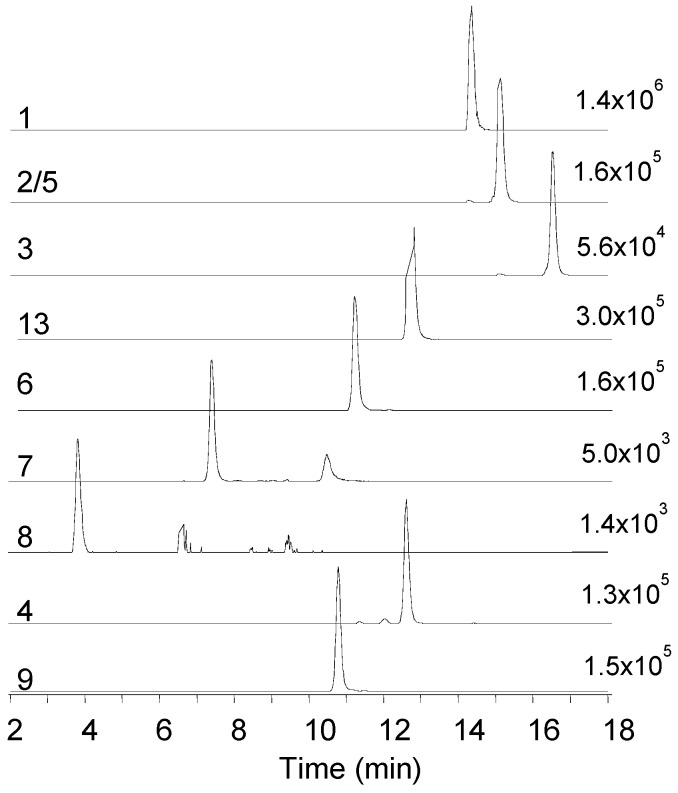
Extracted ion LC–HRMS chromatograms (±5 ppm) of the [M + NH_4_]^+^ ions of beauvericin and beauvenniatin analogues in the crude extract from rice cultures of *Fusarium concentricum* and *Isaria farinosa*; **1**—beauvericin (*m/z* 801.4427), **2/5**—beauvericin A/F (*m/z* 815.4582), **3**—beauvericin C (*m/z* 843.4897), **13**—beauvenniatin L (*m/z* 767.4583), **6**—beauvericin J (*m/z* 817.4409), **7**—beauvericin K (*m/z* 833.4332), **8**—beauvericin L (*m/z* 849.4281), **4**—beauvericin D (*m/z* 787.4269), **9**—beauvenniatin A (*m/z* 753.4432).

**Figure 3 toxins-11-00182-f003:**
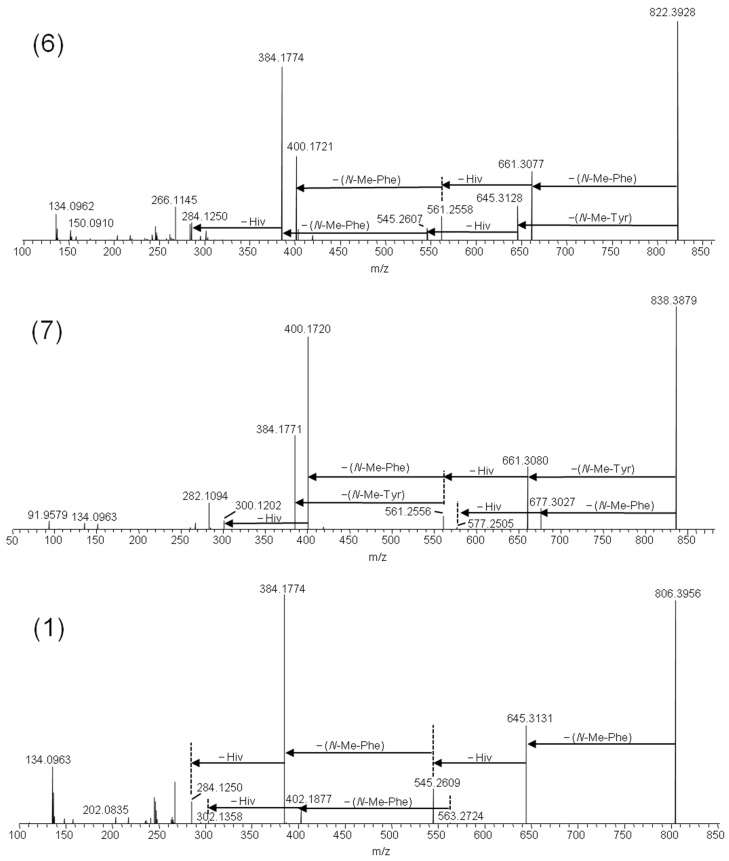
LC–HRMS/MS spectra from higher-collision dissociation of the [M + Na]^+^ ions of beauvericin analogues containing *N*-methyl-tyrosine: beauvericin J (**6**) and beauvericin K (**7**), compared to beauvericin (**1**).

**Figure 4 toxins-11-00182-f004:**
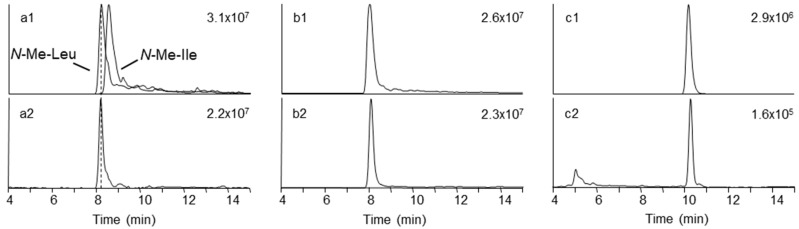
Extracted ion chromatograms from amino acid analysis of hydrolyzed peptide mixtures containing either *N*-methyl-tyrosine containing beauvericins or beauvenniatin L using hydrophilic interaction chromatography ion trap mass spectrometry. Upper traces represent chromatograms from pure reference standards, while lower traces are from hydrolyzed depsipeptide mixtures: (**a1**, **a2**) *N*-methyl-leucine and *N*-methyl-isoleucine; (**b1**, **b2**) *N*-methyl–phenylalanine; and (**c1**, **c2**) *N*-methyl–tyrosine. Individual chromatograms are scaled to the highest peak (number in the top right-hand corner of each chromatogram).

**Figure 5 toxins-11-00182-f005:**
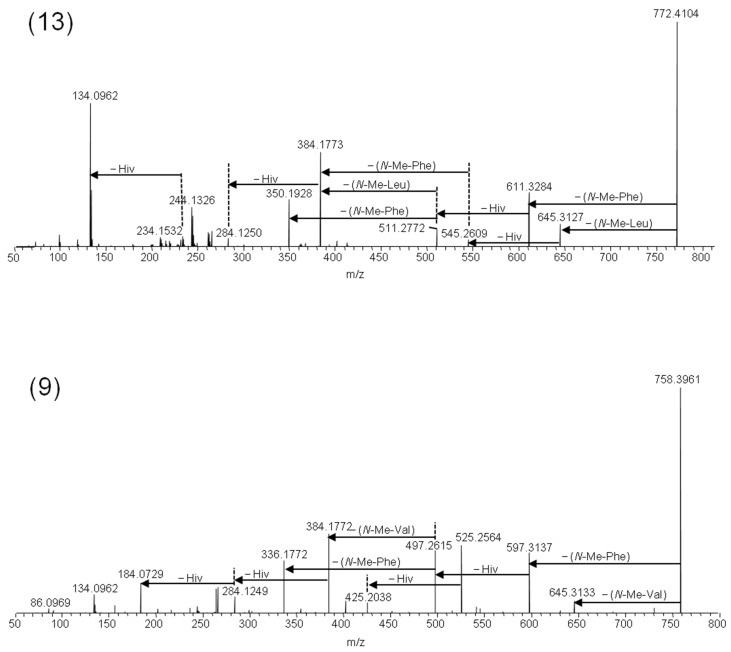
LC–HRMS/MS spectra from the higher-energy collision dissociation of the [M + Na]^+^ ions of beauvenniatin L (**13**) containing *N*-methyl-leucine, and beauvenniatin A (**9**) containing *N*-methyl-valine.

**Table 1 toxins-11-00182-t001:** Fungal strains that were studied in depth for the production of beauvericin and beauvenniatin analogues. Host species, fungal strains identification based on internal transcribed spacers ITS1-ITS2 or translation elongation factor 1α gene (tef-1α) sequence analysis and comparison with reference sequences from the GenBank Database. Beauvericin and beauvenniatin analogues are identified relative to their production rates.

Strain	Species	Host	Sequence Nucleotide Identity	Metabolic Profile *
RT 6.7	*Fusarium proliferatum*	rice (*Oryza sativa*)	99.78% identity to the *Fusarium proliferatum* acc. number JF740730.1	beauvericin (87.0%), beauvericin A/F (0.8%), beauvericin D (3.5%), beauvenniatin A (0.8%), beauvenniatin L (3.2%), beauvericin J (4.7%), beauvericin K (trace)
RT 5.4	*Fusarium proliferatum*	rice (*Oryza sativa*)	99.78% identity to the *Fusarium proliferatum* acc. number JF740730.1	beauvericin (95.6%), beauvericin D (2.6%), beauvenniatin L (0.8%), beauvericin J (0.9%)
MU12	*Fusarium verticillioides*	banana (*Musa* L.)	98.66% identity to the *Fusarium verticillioides* acc. number JF740717.1	beauvericin (18.4%), beauvericin D (8.5%), beauvenniatin A (61.3%), beauvericin K (11.8%)
P35	*Fusarium concentricum*	pineapple (*Ananas comosus*)	100% identity to the *Fusarium concentricum*, acc. number JF740760.1	beauvericin (77.9%), beauvericin A/F (2.2%), beauvericin D (3.6%), beauvenniatin A (3.6%), beauvenniatin L (8.6%), beauvericin J (4.1%), beauvericin K (trace), beauvericin L (trace)
PIN 5.5	*Fusarium proliferatum*	unknown	99.32% identity to the *Fusarium proliferatum* acc. number JF740730.1	beauvericin (87.7%), beauvericin A/F (1.0%), beauvericin D (3.1%), beauvenniatin A (1.2%), beauvenniatin L (5.3%), beauvericin J (1.8%)
4447	*Isaria farinosa*	bark beetle (*Trypodendron lineatum*)	100% identity to the *Isaria farinosa*, acc. number AY624181.1	beauvericin (35.4%), beauvericin A/F (42.5%), beauvericin C (13.8%), beauvericin D (1.5%), beauvenniatin A (0.4%), beauvenniatin L (0.3%), beauvericin J (5.5%), beauvericin K (0.2%), beauvericin L (0.3%)

* The relative peak areas ([M + NH_4_]^+^) of individual analogues as fractions of the total peak area of detected cyclodepsipeptides is shown in brackets.

**Table 2 toxins-11-00182-t002:** Accurate mass and elemental composition of major ions observed for different beauvericin and beauvenniatin analogues from LC–HRMS.

Compound	Measured (*m*/*z*) [M + NH_4_]^+^	Measured (*m*/*z*) [M + Na]^+^	Retention Time (min)	Elemental Composition of Neutral Molecule	Mass Error (ppm) [M + NH_4_]^+^	Mass Error (ppm) [M + Na]^+^
**1** Beauvericin	801.4427	806.3956	14.4	C_45_H_57_N_3_O_9_	0.0	−1.5
**2/5** Beauvericin A/F	815.4582	820.4112	15.1	C_46_H_59_N_3_O_9_	−1.1	−1.7
**3** Beauvericin C	843.4897	848.4433	16.5	C_48_H_63_N_3_O_9_	−0.4	−1.6
**4** Beauvericin D	787.4269	792.3807	12.6	C_44_H_55_N_3_O_9_	−1.2	−2.4
**6** Beauvericin J	817.4409	822.3928	11.1	C_45_H_57_N_3_O_10_	2.6	−1.0
**7** Beauvericin K	833.4332	838.3879	7.4	C_45_H_57_N_3_O_11_	−0.3	−0.8
**8** Beauvericin L	849.4281	854.3825	3.8	C_45_H_57_N_3_O_12_	−0.0	−1.1
**9** Beauvenniatin A	753.4432	758.3961	10.8	C_41_H_57_N_3_O_9_	−0.3	−0.4
**13** Beauvenniatin L	767.4583	772.4104	12.7	C_42_H_59_N_3_O_9_	−0.7	−0.8

**Table 3 toxins-11-00182-t003:** Chromatographic fractionation of crude *Fusarium concentricum* and *Isaria farinosa* extract on silica gel using a chloroform/methanol gradient.

% Methanol	Beauvericin Analogue
1	beauvenniatin A
2	beauvericin, beauvericin A/F, beauvenniatin L, beauvericin C, beauvericin D
4	beauvericin J, beauvericin K, beauvericin L

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
