# Peer review of "Evidence for Naturally Produced Beauvericins Containing N-Methyl-Tyrosine in Hypocreales Fungi"

_toxins, 2019, doi:10.3390/toxins11030182_

Round 1
Reviewer 1 Report
Title: Evidence for naturally produced beauvericins 2 containing N-methyl-tyrosine in Hypocreales fungi
Recommendation: Accept after minor revisions
The current manuscript titled “Evidence for naturally produced beauvericins 2 containing N-methyl-tyrosine in Hypocreales fungi” gives an insight into the natural production of beauvericin analogues among different Hypocreales fungi, such as Fusarium and Paecilomyces spp. The manuscript describes how that N-methyl-tyrosine containing beauvericins may be among the major naturally-produced analogues in certain fungal strain. This is an important field of study as these results provide some theoretical basis for the production of several beauvericin analogues. The results described in the manuscript are significant. However, the outline of the manuscript needs some revision before it can accepted in its current form by the journal.
Major comments:
The manuscript needs to have a separate discussion section. In the current form the authors have merged the discussion section with the individual results section. A separate section will help to compare this study with other studies in the same field. Additionally, it will highlight the significance of the results outlined in this study.
A number of references are missing in the second paragraph of the introduction section. Please include the references in the sections below:
“Beauvenniatins contain one to two amino acids that are commonly found in enniatins, e.g. N- methyl-valine, N-methyl-leucine or N-methyl-isoleucine, in addition to N-methyl-phenylalanine (REF). Beauvenniatins A-F, G1, G2, G3, H1, H2 and H3 were detected in an Acremonium sp. mycelial extract. Beauvenniatins F, G1, G2, G3, H1, H2 and H3 contain one N-methyl-L-phenylalanine and two N-methyl- L-valine residues, the difference between the analogues being variations in their hydroxy acid composition (REF). Several beauvenniatins have only been shown as products from precursor-directed biosynthesis, i.e. addition of the corresponding amino acid precursor to the growing medium was necessary for their production (REF).
Few key references are missing for the manuscript discussion section. The references need to added when a separate discussion section is added in the manuscript (Currently only 23 references).
Author Response
Point 1: The manuscript needs to have a separate discussion section. In the current form the authors have merged the discussion section with the individual results section. A separate section will help to compare this study with other studies in the same field. Additionally, it will highlight the significance of the results outlined in this study.
Response 1: Our understanding is that Toxins allows for a combined Results and Discussion section. A combination of the two sections is especially useful when instrumental data, e.g. MS or NMR data, are presented. The presentation of such data is more difficult to follow without direct discussion, and as our manuscript is primarily about data from instrumental analyses we would like to keep the combined Results and Discussion section.
Point 2: A number of references are missing in the second paragraph of the introduction section. Please include the references in the sections below:
“Beauvenniatins contain one to two amino acids that are commonly found in enniatins, e.g. N- methyl-valine, N-methyl-leucine or N-methyl-isoleucine, in addition to N-methyl-phenylalanine (REF). Beauvenniatins A-F, G1, G2, G3, H1, H2 and H3 were detected in an Acremonium sp. mycelial extract. Beauvenniatins F, G1, G2, G3, H1, H2 and H3 contain one N-methyl-L-phenylalanine and two N-methyl- L-valine residues, the difference between the analogues being variations in their hydroxy acid composition (REF). Several beauvenniatins have only been shown as products from precursor-directed biosynthesis, i.e. addition of the corresponding amino acid precursor to the growing medium was necessary for their production (REF).
Few key references are missing for the manuscript discussion section. The references need to added when a separate discussion section is added in the manuscript (Currently only 23 references).
Response 2: References were added in the mentioned section. The Reviewer did not mention which articles are missing in the Results and Discussion section of the manuscript. This is a publication about a narrow range of research and the authors have added all the publications concerning the subject according to their best knowledge.
Reviewer 2 Report
The authors mainly describe the MS/MS analysis of an important group of fungal natural products. So the subject is interesting, and deserves publication.
However, there are several issues that should be improved prior publication:
Table
1 shows the production of the metabolites by different fungi. Please
include relative production rates to indicate major vs. minor
metabolites.
Of course it would be better tp present NMR data of the new compound.
To improve the usefulness for other researchers, MS fragmentation should be shown for all compounds in the supporting information.
Figure 5 is unclear since there is overlay of arrows and text.
For the biological part:
1. The TEF1 marker is in general suitable for identification of Fusarium spp., (in contrast to ITS, which can only be used to narrow down the species group) but the authors should by all means cite the original literature and disclose the closest matches, and they should also compare the DNA sequences of the type strains because GenBank is full of misidentified sequences of Fusarium isolates.
2. Paecilomyces farinosus belongs to the Eurotiomycetes and it represents asexual state of Byssochlamys. The peptides were hitherto only obtained from those Paecilomyces species that were retained in the Hypocreales as they represent the asexual states of Cordycipitaceae, including Isaria. The authors should therefore carefully double-check the taxonomy of the isolate and if they find that it belongs to the Hypocreales, apply the valid name Isaria rather than Paecilomyces.
Author Response
Point 1: Table 1 shows the production of the metabolites by different fungi. Please include relative production rates to indicate major vs. minor metabolites.
Response 1: We have included relative production rates in Table 1. Therefore the name of Table 1 was changed to "Table 1. Fungal strains that were studied in-depth for production of beauvericin and beauvenniatin analogues. Host species, fungal strains identification based on ITS1-ITS2 or tef-1α sequence analysis and comparison with reference sequences from GeneBank Database. Beauvericin and beauvenniatin analogues relative production rates."
Point 2: Of course it would be better tp present NMR data of the new compound.
Response 2: We agree that NMR data would be necessary to unequivocally identify the new compound, and such data would also be good for verification of some of the other major analogues. The study would then however become something different, and also require resources that were not available. Our idea was to show how to study the structural variability of these peptides with instrumentation that is nowadays widely available.
Point 3: To improve the usefulness for other researchers, MS fragmentation should be shown for all compounds in the supporting information.
Response 3: MS fragmentation was shown in manuscript and supplementary data for ten compounds which are also characterized in Figure 2 as LC–HRMS chromatograms. Three compounds: beauvenniatin C, D and E were only discussed in manuscript. Therefore the name of Figure 1 was changed to " Figure 1. Overview of chemical structures of beauvericin and beauvenniatin analogues discussed in this study."
Point 4: Figure 5 is unclear since there is overlay of arrows and text.
Response 4: Figure 3 and Figure 5 were improved and replaced in the revised manuscript.
Point 5: The TEF1 marker is in general suitable for identification of Fusarium spp., (in contrast to ITS, which can only be used to narrow down the species group) but the authors should by all means cite the original literature and disclose the closest matches, and they should also compare the DNA sequences of the type strains because GenBank is full of misidentified sequences of Fusarium isolates.
Response 5: Original literature were cited and sequences were compared to reference genes from GenBank Database in Table 1. We have made new molecular analyses and compared to the previous results. Results have shown that RT 6.7 strain was re-identified as Fusarium proliferatum and PIN 5.5 strain was also re-identified as Fusarium proliferatum. The differences between new and old results in PIN 5.5 strain identification may result from contamination of this strain during 6 years of storage. Also incorrect annotation of the reference strain in the GenBank Database is also possible. Here, we have cleared the identity of the strain. The differences between new and old results in RT 6.7 strain identification follow from incorrectly identified strain in previous screening works.
Point 6: Paecilomyces farinosus belongs to the Eurotiomycetes and it represents asexual state of Byssochlamys. The peptides were hitherto only obtained from those Paecilomyces species that were retained in the Hypocreales as they represent the asexual states of Cordycipitaceae, including Isaria. The authors should therefore carefully double-check the taxonomy of the isolate and if they find that it belongs to the Hypocreales, apply the valid name Isaria rather than Paecilomyces.
Response 6: Paecilomyces farinosus was changed to Isaria farinosa.
Round 2
Reviewer 2 Report
The authors improved the manuscript significantly. Please round the metabolic profile values (i.e. (0.795%) to sensibly (i.e. 0.8 %).
Author Response
Point 1: The authors improved the manuscript significantly. Please round the metabolic profile values (i.e. (0.795%) to sensibly (i.e. 0.8 %).
Response 1: The metabolic profile values in Table 1 were changed